# Improvements to a laser-induced fluorescence instrument for measuring SO₂: impact on accuracy and precision

Pamela S. Rickly[1,2], Lu Xu[3], John D. Crounse[3], Paul O. Wennberg[3,4], and Andrew W. Rollins[2]

[1]Cooperative Institute for Research in Environmental Sciences, University of Colorado, Boulder, CO 80309, USA

[2]Chemical Sciences Laboratory, National Oceanic and Atmospheric Administration, Boulder, CO 80305, USA

[3]Division of Geological and Planetary Sciences, California Institute of Technology, Pasadena, CA 91125, USA

[4]Division of Engineering and Applied Science, California Institute of Technology, Pasadena, CA 91125, USA

**Abstract.** This work describes key improvements made to the *in-situ* laser induced fluorescence instrument for measuring sulfur dioxide ($SO_2$) that was originally described by Rollins et al. (2016). Here we report measurements of the $SO_2$ fluorescence emission spectrum. These measurements allow for the determination of the most appropriate bandpass filters to optimize the fluorescence signal while reducing the instrumental background. Because many aromatic species fluoresce in the same spectral region as $SO_2$, fluorescence spectra were also measured for naphthalene and anisole to determine if ambient $SO_2$ measurements could be biased in the presence of such species. Improvement in the laser system resulted in better tunability, and a significant reduction in the 216.9 nm laser linewidth. This increases the online/offline signal ratio which in-turn improves the precision and specificity of the measurement. The effects of these improvements on the instrumental sensitivity were determined by analyzing the signal and background of the instrument using varying optical bandpass filter ranges and cell pressures and calculating the resulting limit of detection. As a result, we report an improvement to the instrumental sensitivity by as much as 50%.

## 1 Background

Sulfur dioxide ($SO_2$) is responsible for a number of health and environmental impacts. Through reaction with the hydroxyl radical (OH), $SO_2$ produces sulfuric acid which affects the pH of aqueous particles and leads to acid deposition. Sulfuric acid also condenses onto organic and black carbon particles producing sulfate which increases the aerosol hygroscopicity and influences the accumulation of aerosol liquid water (Fiedler et al., 2011; Carlton et al., 2020). Sulfuric acid is believed to be the most important source gas globally for homogeneous nucleation and growth of new aerosol particles, which may occur primarily in the tropical upper troposphere (Brock et al., 1995; Dunne et al., 2016; Williamson et al. 2019). $SO_2$ and sulfate particles can be transported long distances driving the production of haze pollution in areas downwind of $SO_2$ emissions (Andreae et al., 1988). Both the direct radiative forcing from aerosol and the indirect forcing from aerosol cloud interactions are important for climate. While both tend to produce an offset to greenhouse gas induced warming by reducing incoming shortwave radiation, the effect of aerosol cloud interactions is complicated and produces large uncertainties in climate models (Finlayson-Pitts and Pitts, 1999; IPCC, 2018). Changing emissions distributions coupled with incomplete understanding of the chemistry and microphysics associated with sulfur and aerosol formation in the atmosphere necessitates further studies which require precise and accurate measurements of $SO_2$ throughout the troposphere and lower stratosphere.

Regulation of anthropogenic emissions has resulted in decreased atmospheric $SO_2$ concentrations in the United States and Europe since the 1970's. However, during the early 21st century, emissions began increasing dramatically in Asia as a result of increased fossil fuel burning (Smith et al., 2011; Hoesly et al., 2018). The main source of $SO_2$ to the troposphere is through direct emission followed by oxidation of DMS. The remaining pathways of $SO_2$ formation, $H_2S$, $CS_2$, and OCS oxidation, contribute negligible fluxes (Feinberg et al., 2019). As of 2014, global emission rates of $SO_2$ were reported to be approximately 113 Tg S yr$^{-1}$ - more than double the flux during the 1950's (Hoesly, et al., 2018). Anthropogenic sources of sulfur, mainly from fossil fuel combustion and smelting, are the largest global sources of $SO_2$ to the atmosphere and as of year 2000 comprised around 67% of total global $SO_2$ emissions (Feinberg et al., 2019; Lee et al., 2011; Smith et al., 2011). Biogenic sources make up a small, but significant fraction of $SO_2$ emissions with the majority derived from marine phytoplankton, in the form of

dimethyl sulfide which exhibits a global source rate that is approximately 26% of total global $SO_2$ input (Lee et al., 2011, Feinberg et al., 2019).

Although global emission rates of $SO_2$ have continued to decrease since the early 21[st] century, atmospheric sulfur concentrations are expected to be affected by continued climate change and could represent feedback mechanisms within the climate system. Due to reductions of sulfur deposition, Hinckley et al. (2020) has found that farmers in the United States are needing to apply sulfur containing fertilizer to croplands to enhance nitrogen uptake to plants at a rate of 20-300 kg S $yr^{-1}$. In addition, Kesselmeier et al. (1993) have reported that terrestrial sources of sulfur exhibit behavior similar to monoterpenes in that they are light and temperature dependent. This suggests that increasing sulfur emissions are likely to occur as global temperatures continue to rise. Lastly, the effect of climate warming on the variability of moisture conditions, as well as increased land use change, is expected to increase both the frequency and duration of biomass burning events which is expected to further increase sulfur emissions (Westerling et al., 1990; Heyerdahl et al., 2002). In combination, while fossil fuel burning sources of $SO_2$ are continuing to decrease, it is likely that these climate-related and additive sources may keep $SO_2$ emissions from returning to pre-industrial levels.

Even small $SO_2$ mixing ratios can produce important effects. Remote regions, including much of the equatorial marine boundary layer, exhibit mixing ratios of $SO_2$ on the order of 100 ppt. Still, in these regions, the biogenic $SO_2$ may be the primary source of cloud condensation nuclei. In addition, convective transport from these regions into the tropical tropopause layer can allow these small sources to reach the lower stratosphere. Sulfate aerosol lifetimes in the stratosphere are approximately 100 times that of aerosol within the lower troposphere allowing them to persist for 1-2 years (Holton et al., 1995). As a result, sources of sulfate aerosol and aerosol precursor species reaching the UT/LS are disproportionately important for climate compared to short lived aerosols in the lower troposphere. However, to date, few studies have reported measurements of $SO_2$ in the UT/LS (Inn and Veder, 1981; Georgii and Meixner, 1980; Rollins et al., 2017, 2018). Understanding in detail the impact that $SO_2$ has on the stratosphere is only becoming increasingly important as discussions of albedo modification by injection of $SO_2$ into the stratosphere is becoming more common (National Research Council, 2015).

Despite the potential implications that changing $SO_2$ concentrations present in the UT/LS and the remote lower troposphere, few in situ measurements are routinely made in either of these areas. Most direct measurements have been made through the use of pulsed fluorescence instruments, which are available commercially; however, this technique tends to exhibit interferences from other fluorescent species and limited precision. Most measurements of $SO_2$ with pptv precision have been made using chemical ionization mass spectrometry (CIMS). Many CIMS $SO_2$ ionization chemistry schemes can be sensitive to variations in ambient water vapor, complicating tropospheric measurements (Huey et al., 2004; Eger et al., 2019). Operation of CIMS instruments on unpressurized aircraft capable of reaching the tropical lower stratosphere (>17 km) is also challenging from an engineering perspective.

As an alternative, the development of a compact *in situ* laser induced fluorescence (LIF) based $SO_2$ instrument was recently reported by Rollins et al. (2016) with a $1\sigma$ precision of 2 ppt over a 10 s integration period. That technique was developed and originally used to quantify $SO_2$ in the UT/LS region on two NASA WB-57F missions; VIRGAS (Volcano Investigation Readiness and Gas and Aerosol Sulfur) in 2015 and POSIDON (Pacific Oxidants Sulfur Ice and DehydratiON) in 2016. In the UT/LS where potentially interfering fluorescent species (e.g. aromatic compounds) are in negligible concentrations, the instrument was used in a mode where the excitation laser was maintained on an $SO_2$ resonance and the optical background was determined using periodic $SO_2$-free zero-air additions. However, it is expected that LIF measurements of $SO_2$ in the more complex chemical environment of the troposphere, and especially in areas where fossil fuel combustion is occurring and through biomass burning plumes might result in interferences from species including aromatics that are also formed during these processes. Prior to the deployment of the instrument on a NASA Global Hawk mission in 2017 (HOPE-EPOCH), the LIF instrument was improved to allow for rapid dithering of the excitation laser on and off of an $SO_2$ resonance, to allow for continuous discrimination of $SO_2$ from other fluorescent species. It was also operated this way during the NASA ATom-4 mission (Atmospheric Tomography Mission) in 2018 and the NASA/NOAA Fire Influence on Regional to Global Environments Experiment – Air Quality experiment in 2019 (FIREX-AQ). Following ATom-4, it has been investigated how further improvements to the detection of $SO_2$ might be accomplished by quantifying the spectral region of $SO_2$ fluorescence for separation from scattering and identification of fluorescence emissions from potentially interfering aromatic species. Here we report on these recent improvements and use of the instrument.

**2 Laser induced fluorescence**

**2.1 $SO_2$ spectroscopy**


Fluorescence occurs during the radiative relaxation of a molecule after its absorption of a photon. Thus, the LIF signal is proportional to the molecular absorption cross section at the laser wavelength and the quantum yield for fluorescence. A study by Manatt and Lane (1993) compiled numerous measurements of $SO_2$ absorption cross-sections over varying wavelengths to find six absorption bands between 100 and 400 nm. Because strong absorption

by oxygen ($O_2$) occurs below 200 nm, the three spectral regions between 200 and 400 nm are most appropriate for atmospheric measurements of $SO_2$. Excitation into the $\tilde{C}(^1B_2)$ state with a wavelength region of 170-235 nm, is typically chosen for $SO_2$ fluorescence detection due to the higher absorption cross-section and fluorescence quantum yield compared to longer wavelengths. Pumping into the B or A bands ($\lambda > 240$ nm), while useful for absorption measurements, results in negligible fluorescence. Another consideration is the predissociation threshold near 218.7

nm (Bludsky et al., 2000). As a result, pumping at wavelengths less than ~215 nm results in negligible fluorescence. Considering the absorption cross section and fluorescence quantum yields, excitation at 220.6 nm is theorized to produce the maximum detectable signal using LIF (Rollins et al., 2016).

Due to the limited availability of practical laser technology for airborne instrumentation, Rollins et al. (2016) targeted a band-head at 216.9 nm through the use of the fifth harmonic produced from a tunable 1084.5 nm

fiber-amplified diode laser. The detected fluorescence was selected using a long-pass filter (Thorlabs FGUV5) in combination with a bandpass filter (Asahi XUV0400) allowing transmission of the red-shifted fluorescence in the wavelength region of 240-400 nm. A more detailed description of the LIF $SO_2$ instrument can be found in Rollins et al. (2016). Generally, while precision sufficient for measurements in the UT/LS was achieved (2 ppt over 10 s integration period), the detection limit was primarily controlled by the background from scattered photons, and it

was anticipated that in polluted regions where other fluorescent species exist, the detection limit might be further degraded due to these additional sources of background.

## 2.2 Laser subsystem improvements

The $SO_2$ instrument uses a custom-built fiber laser system. Pulses from a fiber coupled tunable diode laser near 1084.5 nm are used to seed a fiber amplifier system. Originally, the seed laser operated in a gain-switching mode where short pulses of current injected into the seed laser generated the ~ 1 ns optical output pulses. In the present design, the seed laser is instead operated in a constant current mode, and its output is modulated using a fiber-coupled electro-optic modulator (EOM). The EOM can produce pulses of less than 1 ns full width at half maximum,

and with an extinction ratio of 40 dB. While the original design was somewhat simpler to operate than the present design, gain-switching of the seed laser significantly broadened the laser spectrum and eliminated the possibility of tuning the laser wavelength by modulating the seed laser injection current. Rather, the wavelength of the system was tuned by adjusting the temperature of the seed laser, which had a settling time on the order of seconds. The new design is also operated at a laser repetition-rate of 200 kHz instead of the 25 kHz of the original design. This reduces

the peak power leading to less spectral broadening within the fibers and also increases the dynamic range of the single-photon counting LIF signal to 200 kHz rather than 25 kHz. While less UV laser power is achieved with the higher repetition rate and lower pulse energy, the overall sensitivity to $SO_2$ improves because the $SO_2$ spectrum can now be fully resolved which increases the convolution of the $SO_2$ spectrum with the laser spectrum.

Output from the fiber amplifier is passed through three nonlinear crystals (KTP, LBO, BBO) in the same

configuration as described by Rollins et al. (2016) to produce the fifth harmonic of the fiber output at 216.9 nm with typically 1 mW of power. Overall, injecting the seed laser at constant current and operating the amplifier with lower peak powers results in a significantly narrower laser linewidth. Due to the narrower laser linewidth, the Doppler broadened $SO_2$ spectrum can now be fully resolved (Fig. 1). This significantly increases the effective $SO_2$ absorption cross-section. The laser wavelength can also be tuned rapidly to measure on and off of an $SO_2$ resonance many times

each second which eliminates spectral interferences from other fluorescent species (Fig. 2).

## 2.2 Fluorescence background

Signal on the detector may result from $SO_2$ fluorescence, or from a suite of other sources. These include Rayleigh,

Raman, or aerosol scattering, or fluorescence from the LIF chamber or windows, and red-shifted fluorescence from other gases and aerosols in the sample. Because the $SO_2$ absorption spectrum has fine structure, the signal from $SO_2$ can selectively be reduced by more than one order of magnitude by tuning the laser less than 10 pm off of an $SO_2$ resonance (Fig. 1). All other sources of photons, however, are expected to have no appreciable structure at this spectral resolution. Therefore, the signal from $SO_2$ can accurately be distinguished from other photon sources by

constantly tuning the laser on and off of an $SO_2$ resonance.

The instrumental precision, however, is determined by the Poisson counting statistics of the sum of the $SO_2$ fluorescence and background signals which, at low $SO_2$ mixing ratios, is dominated by the non-$SO_2$ count rate. Therefore, the detection limit is determined by these background sources of photons.

Knowledge of the $SO_2$ emission spectrum is key for choosing detection bandpass filters to maximize the $SO_2$ signal while minimizing detection of photons from non-$SO_2$ sources. While the Rayleigh and Raman scatter by $N_2$ and $O_2$ occur at constant and known wavelengths, red-shifted fluorescence from other gases and aerosols may vary by compound. Many aromatic species have been reported to have non-negligible absorption cross-sections and fluorescence quantum yields when pumped near 216.9 nm. Because aromatics and $SO_2$ are co-emitted during combustion of many fuels, it is important to understand the affect they may produce on ambient LIF $SO_2$ measurements.

While many aromatics are released during combustion, two compounds reported as having large emission ratios and significant absorption cross-sections at 216.9 nm are naphthalene and anisole (Grosch et al., 2015; Koss et al., 2018; Mangini et al., 1967; Warneke et al., 2011). Emission ratios of these compounds are dependent on the type of combustion, fossil fuel or biomass, and the elements involved in the combustion. Naphthalene is released through both fossil fuel combustion and biomass burning with the latter producing an emission ratio of greater than 1 ppb/ppm CO (Warneke et al., 2011). Anisole is primarily released through biomass burning with an emission ratio reported in combination with cresol as 1.5 ppb/ppm CO (Koss et al., 2018). The absorption cross-sections reported for these compounds near 216.9 nm are $2.6 \times 10^{-17}$ $cm^2$ molecule$^{-1}$ for naphthalene (Grosch et al., 2015) and $2.1 \times 10^{-17}$ $cm^2$ molecule$^{-1}$ for anisole (Mangini et al., 1967). With the emission ratio of $SO_2$ being similar to these aromatics, in addition to similar absorption cross-sections, it is anticipated that aromatics could significantly interfere with ambient $SO_2$ measurements. To optimize the detection of the fluorescence spectral region for $SO_2$, we measured the spectral distributions of the fluorescence emission from $SO_2$, naphthalene, and anisole.

## 3 Measurement of scattering and fluorescence spectra

The optical bench used for the LIF $SO_2$ detection is shown in Fig. 3. The 216.9 nm laser, approximately 1 mW, enters the cell perpendicular to the entrance of the sampled air. Sampled air (2500 sccm) enters the system through a custom butterfly valve machined from PEEK (polyether ether ketone) that reduces the pressure to 170 mbar which shows minimal change through the sample cell. The majority of the gas exits the sample cell opposite of the inlet, while 250 sccm is exhausted through each of the cell side arms to eliminate dead space in the flow system. Detection of the fluorescence is then measured orthogonally to both the sample flow and laser axes. A UV fused silica aspheric lens (Edmund Optics, NA = 0.5) is used to collect approximately 10% of the solid angle relative to the center of the cell. After passing through the measurement cell, the beam passes through a LIF reference cell with a similar arrangement to the measurement cell. A constant flow with a mixing ratio near 500 ppb $SO_2$ is maintained in the reference cell. Feedback from the reference cell is used to ensure that the laser is tuned to the $SO_2$ resonance peak, and to quantify any changes in the instrument sensitivity due to changes in the laser spectrum. The exhausts of the measurement and reference cells are tied together such that small perturbations in the system pressure will also be equally observed in the two cells. A National Instruments cRIO computer system is used for controlling all timing requirements. This includes the timing of the seed laser pulses, amplification, and photon counting and gating.

For measurements of the fluorescence emission spectra of $SO_2$ and aromatic compounds, the entrance of a round-to-linear fiber optic bundle (ThorLabs FG105UCA) was placed at the focal point of the lens where the fluorescence detecting PMT is typically located. The emission from the linear end of the fiber bundle was focused by a collimating lens (74-series Ocean Optics) onto the entrance slit of a scanning monochromator (Acton Research Corporation VM-502). Measurements were made with the monochromator in the V configuration with slit sizes of 4 mm wide using a Hamamatsu photomultiplier tube (H12386-210) connected at the exit of the monochromator.

In order to measure the $SO_2$ fluorescence spectrum, a flow of 46 sccm from a 5 ppm $SO_2$ gas cylinder (Scott Marrin) was mixed with 2500 sccm zero air producing a concentration of approximately 90 ppb that was sampled into the cell. Figure 4 shows the observed spectrum in the presence (red) and absence (orange) of $SO_2$. The signal observed at 202-236 nm in the absence of $SO_2$ is primarily from Rayleigh scatter of the laser and the observed width is a measure of the spectral resolution of the experimental setup. The figure to the right shows just the $SO_2$ fluorescence calculated after the subtraction of the background.

To verify the calibration and spectral resolution of the monochromator, a low-pressure double-bore mercury (Hg) capillary lamp (Jelight) was positioned in front of the fiber as a spectral reference. The atomic Hg emission spectrum from 200 – 500 nm was measured using the monochromator to calibrate the monochromator and show that the monochromator-fiber setup produces a spectral resolution with a full width-half max (FWHM) of

approximately 20 nm. Figure 4 shows that the $SO_2$ fluorescence emission spectrum appears to occur in two regions. The main peak is centered around 302 nm and produces a FWHM value of 63 nm in our setup. A second peak may exist between 350-360 nm; however, it is not possible to accurately determine the peak position nor the width in this setup. Using the monochromator with relatively wide slits was necessary to achieve enough signal to measure the fluorescence emission spectrum in our setup, and somewhat exaggerates the width of the emission spectrum in figures 4 – 6.

Measurements were similarly made with the aromatic compounds. Zero-air was passed through the head-space of a vial containing a sample of pure (>99%) crystalline naphthalene (Aldrich), which has a vapor pressure of 0.04 mbar at 298K. This resulted in a naphthalene concentration of approximately 3.6 ppm when the flow through the vial was 100 sccm and the total flow through the instrument was 1300 sccm. Liquid anisole (Sigma-Aldrich), with a vapor pressure of 5 mbar at 298K, was similarly used to produce a concentration of approximately 26 ppm when the flow through the vial was 10 sccm and the total flow was 2260 sccm (Ambrose et al., 1976). A comparison of the fluorescence spectra of the aromatic compounds to the $SO_2$ fluorescence spectrum is shown in Fig. 5.

Anisole produces a similar fluorescence spectrum as the main $SO_2$ emission peak with a maximum at 304 nm. The FWHM of 46 nm occurs between 286 and 332 nm, the latter half of the largest $SO_2$ fluorescence peak. Similarly, naphthalene produces a fluorescence spectrum peaking around 340 nm with a FWHM value of 46 nm, which slightly overlaps with the tail end of the largest $SO_2$ fluorescence peak and nearly completely overlaps with the second $SO_2$ region.

Figure 5 shows that aromatic compounds could produce significant signal in the $SO_2$ instrument during measurements in polluted environments. While naphthalene has a similar absorption cross section and rate of collisional quenching as $SO_2$, its low-pressure fluorescence quantum yield has been reported to be 2-3 times larger due to a lower expected rate of photodissociation as a result of a larger dissociation energy threshold (Hui and Rice, 1973; Martinez et al., 2004; Reed and Kass, 2000; Suto et al., 1992). However, the observed naphthalene signal in our experiment is approximately 30 times lower than $SO_2$. Because of its reduced rate of photodissociation, the fluorescence lifetime of naphthalene (340 ns) is much greater than that of $SO_2$ (30 ns) (Hui and Rice, 1973; Martinez et al., 2004). Therefore, the short counting gate used in this work for $SO_2$ detection (25 ns) discriminates the majority of the naphthalene signal from that of $SO_2$.

No difference in the fluorescence emission (intensity or spectral distribution) was observed for the aromatic compounds with the laser tuned on or off of the $SO_2$ resonance. This is consistent with expectations based on the literature absorption cross sections for those compounds showing no fine structure comparable to the features in the $SO_2$ spectrum (Keller-Rudek et al., 2013). Therefore, these compounds would only increase the instrument background resulting in reduced precision, but would result in unbiased ambient $SO_2$ measurements in areas of high aromatic concentrations.

**4 Implementation of bandpass filters**

In order to achieve the highest sensitivity and lowest limit of detection, the impact of the fluorescence detection bandpass filters was further investigated. With the instrument in its normal configuration for measuring $SO_2$ (PMT photocathode located at focal point of fluorescence collection lens), different bandpass filters were used in front of the detection cell PMT to directly measure the $SO_2$ fluorescence signal and quantify the background scatter over a few discrete regions of the spectrum. Figure 6 shows the fluorescence spectrum (red) and background (orange) observed using the monochromator scaled to the count rates observed with each of the tested filters. It was observed that the background increases with wavelength until around 300 nm. After this point, the background slowly decreases reaching a minimum around 400 nm before increasing again near 420 nm. The boxes and closed circles are indicative of the filter measurements. The heights of the boxes indicate the fluorescence signal observed and the width shows the range in which transmission was achievable with the filter. The closed circles indicate the background observed with the filter of the corresponding color.

To optimize the $SO_2$ detection limit in zero air, the detection limit was calculated from the scaled $SO_2$ fluorescence and background spectra using a theoretical bandpass filter assuming a low pass of 246 nm and a variable high pass limit where 100% transmission is observed in the pass band and 0% transmission elsewhere. The detection limit was calculated with the filter high pass limit at 270 nm and in increasing increments of 10 nm to the full spectrum at 500 nm. Figure 7 shows the results of this calculation as a function of the high pass filter limit. As the upper end of the pass band is initially increased from 250 to 300 nm, the calculated detection limit rapidly decreases due to significant gains in signal relative to background here. A minimum in the $SO_2$ detection limit occurs over the range of 350-450 nm with a theoretical detection limit as low as 1.8 ppt over a 1 second integration

period. For polluted environments where 1 ppb of naphthalene may be present, the detection limit would increase by 20% within this detection range due to the increased background from naphthalene fluorescence.

The dielectric bandpass filter that we have used previously for $SO_2$ measurements in the atmosphere (Asahi XUV0400) efficiently reduces signal from Rayleigh scatter, and we find that the additional inclusion of the absorptive glass filter (Thorlabs FGUV5) originally used with this instrument is no longer necessary. Removal of the FGUV5 filter allows for greater transmission within the $SO_2$ fluorescence spectrum range and provides the largest signal to noise ratio of the sampled bandpass filters. However, the transmission is still limited to 80-95% with the Asahi XUV0400 filter alone in comparison to the theoretical filter. This results in a detection limit of approximately 3.4 ppt over a 1 second integration period, as shown by the marker in Fig. 7. Testing showed that, the detection limit decreases with increasing pressure up to at least 250 mbar (Fig. 8). Because predissociation limits the $SO_2$ fluorescence lifetime to ~5 ns, the fluorescence quantum yield is rather insensitive to pressure in this regime, and therefore the signal increases linearly with pressure due to the increased number density of $SO_2$. While these measurements were performed under dry conditions, ambient measurements are expected to produce similar results because the short fluorescence lifetime will also limit the importance of collisional quenching by water vapor (Rollins et al., 2016). As a result, the detection limit can be reduced by approximately 15% by increasing the cell pressure to around 250 mbar. While this is beneficial for measurements in the lower troposphere, UT/LS measurements will require the cell pressure to remain less than 100 mbar in order for the instrument to maintain an adequate flow rate.

## 5 Effect of Ozone

Typical mixing ratios of ozone that can be encountered during stratospheric sampling are not anticipated to affect the $SO_2$ LIF signal. However, due to the relatively high quantum yield for $SO_2$ photolysis at 216.9 nm forming primarily sulfur monoxide (SO), the possibility of significantly enhancing the LIF signal through the chemiluminescent reaction of SO with ozone was investigated using higher ozone mixing ratios (Hui and Rice, 1972; Okabe et al., 1971; Ryerson et al., 1994). Figure 9 shows the observed LIF signal in the presence of significant additions of ozone to the inlet. At ozone additions up to 2000 ppm, small increases in LIF signal were observed with increasing $O_3$ (2 % LIF signal / ppm $O_3$). This further demonstrates that at stratospheric $O_3$ mixing ratios accessible by aircraft (< 5 ppm), the signal changes by < 10 %. At $O_3$ above 2000 ppm, significant decreases in the LIF signal were observed. We attribute these decreases to photolysis of $O_3$ at 216.9 nm followed by destruction of $SO_2$ by fast reaction (2.2 $\times 10^{-10}$ $cm^3$ molecule$^{-1}$ s$^{-1}$) with the atomic oxygen produced by $O_3$ photolysis (Sander et al., 2011).

## 6 Field performance

With the instrument configured with the typical bandpass filter (Asahi XUV0400) and at a pressure of 170 mbar, calibrations were performed to assess the impact of the improvements on the precision of the $SO_2$ measurement. Calibrations of the instrument are performed in which a mixture of zero air and $SO_2$ standard are introduced to the instrument with a mixing ratio range of around 1.6-8 ppb $SO_2$. This mixture is comprised of a flow of 1-5 sccm of 5 ppm $SO_2$ with 3000 sccm zero air which has passed through a $KMnO_4$ scrubber removing any $SO_2$ that may be present in the zero air.

An example of the new calibration is shown in Fig. 10. Typical sensitivity is 26 counts per second (CPS) ppt$^{-1}$ for $SO_2$ and 1000 CPS of background. Thus, the background is a photon count rate equivalent to 38 ppt of $SO_2$. In our previous work (Rollins et al. 2016) we reported an in-flight background of 480 CPS and sensitivity of 4.1 CPS ppt$^{-1}$, or a background equivalent to 117 ppt of $SO_2$. Therefore, the signal relative to background has increased threefold, which we attribute primarily to the narrower laser linewidth in the new configuration. With the new signal level, the $1\sigma$ detection limit for 1 Hz measurements is 3.4 ppt. For 10 seconds of integration time the detection limit would be 1.1 ppt, which is nearly half of what we previously stated for 10 s of integration.

During the NASA ATom-4 field campaign, measurements of $SO_2$ were acquired by both the LIF instrument as well as the California Institute of Technology CIMS instrument (CIT CIMS). ATom-4 sampled primarily pristine air masses with a limited number of measurements of ship emissions, biomass burning plumes, and volcanic emissions. This allowed for the first in-situ comparison between the current LIF technique and another $SO_2$ measurement method. CIT CIMS uses fluoride ion transfer chemistry from $CF_3O^-$ reagent ion (e.g. $SO_2 + CF_3O^- \rightarrow SO_2 \cdot F^- + CF_2O$) followed by mass spectral analysis using a compact time of flight mass spectrometer (CToF) with typical mass resolution of $m/\Delta m = 1,200$. The precision of CIMS measurements degrades with increasing water vapor concentration because of rising interference of formic acid signal ($CH_2O_2 \cdot H_2O \cdot CF_3O^-$) which has a mass that

differs from $SO_2$ by only 0.054 Da. In the marine boundary layer when water vapor was $> 20 \times 10^3$ ppm, the CIMS $SO_2$ precision ($1\sigma$ standard deviation over a 1 s integration period) is larger than 130ppt, a value greater than the typical $SO_2$ concentration (<100 pptv) as reported by the LIF instrument. Therefore, CIMS measurements when water vapor was $> 20 \times 10^3$ ppm are excluded from the comparison. Figure 11a shows an orthogonal regression of both measurements from ATom-4 at 10-second time resolution. Overall, an excellent correlation was observed between the two instruments ($R^2 = 0.99$). Furthermore, measurements during FIREX-AQ acquired during smoke plume penetration again show excellent correlation between the CIT CIMS and SO2 LIF instruments (Fig. 11b). This indicates that aromatic compounds are not biasing the SO2 LIF measurements. The CIMS instrument reported 12% and 10% lower $SO_2$ than LIF during the ATom-4 and FIREX-AQ missions, respectively. While these are within the combined uncertainties of the measurements, it indicates a systematic calibration error with one or both of the instruments that has not been resolved at this time.

**7 Summary**

Rollins et al. (2016) reported a new *in situ* method of measuring $SO_2$ in the UT/LS using LIF. Here we report improvements to the technique that allow for measurements in polluted areas containing other fluorescent species and an overall reduction in the detection limit. This was accomplished by limiting non-$SO_2$ fluorescence background sources and by improvements in the linewidth and tunability of the laser system. Similar to $SO_2$, aromatic species are largely emitted during combustion processes, many of which have large absorption cross-sections near 216.9 nm and significant fluorescence quantum yields. To determine the effect of these compounds on measuring $SO_2$, the fluorescence spectra of $SO_2$ and two aromatic compounds, naphthalene and anisole, were measured. While strong overlap was exhibited in the fluorescence spectra of these aromatic species with $SO_2$, the excitation spectrum of $SO_2$ has fine structure near 216.9 nm while the excitation spectra of the aromatics is relatively invariant near 216.9 nm. Therefore, these compounds will only increase the background, slightly reducing the precision of the instrument, but will not result in biased $SO_2$ measurements. Similar consequences on the LIF $SO_2$ measurements are expected from other aromatic species since those species generally do not have fine structure in their excitation spectra.

       The $SO_2$ fluorescence spectrum was also used to determine the bandpass filter application that would best limit the amount of scatter observed and provide the best limit of detection. Using a theoretical bandpass filter with transmission beginning at 246 nm, the ending transmission range in which the lowest detection limit is expected is 350-450 nm. This is expected to result in a detection limit of 1.8 ppt. Using the Asahi XUV0400 bandpass filter, we are able to reach a detection limit of 3.4 ppt.

       Improvements in the laser system are responsible for significant performance improvements over our previous work. Here, we reduced the laser linewidth which increased the LIF signal by nearly a factor of three. The laser wavelength is now controlled by current tuning of the laser, which can be performed rapidly allowing for measurements of online and offline fluorescence signals many times in a second. This eliminates the possibility of spectral interferences from other species such as aromatics. Increasing the laser repetition rate from 25 kHz to 200 kHz also greatly increased the dynamic range.

       While it is shown that increasing the cell pressure to around 250 mbar would reduce the limit of detection by approximately 15%, this higher cell pressure can only be used in the lower troposphere. Measurements in the UT/LS will require the cell pressure to be operated near 40-50 mbar. It was also demonstrated that increased ozone concentrations observed in the UT/LS would not significantly influence LIF $SO_2$ measurements. The resulting production of $O(^1D)$ does not become large enough to decrease $SO_2$ within the sampling cell until reaching ozone concentrations greater than 4000 ppm, more than two orders of magnitude greater than stratospheric ozone concentrations.

       The culmination of these changes to the LIF $SO_2$ instrument has resulted in an increased instrumental sensitivity and lower limit of detection. Calibrations suggest that the instrumental sensitivity has improved by approximately 3x relative to the background from that reported in Rollins et al. (2016). Comparison with measurements performed by the CIT CIMS instrument during the NASA ATom-4 and FIREX-AQ campaigns demonstrated good agreement. These results suggest that the LIF $SO_2$ instrument is highly suitable for measurements in both polluted and pristine environments.

*Data availability.* The data collected for FIREX-AQ are available from the NASA/NOAA FIREX-AQ data archive: https://www-air.larc.nasa.gov/cgi-bin/ArcView/firexaq. The data collected for ATom-4 are available from the NASA ESPO data archive: https://espoarchive.nasa.gov/archive/browse/atom/id14.

*Author contribution.* The research was designed by PSR and AWR.  Measurements were taken by PSR, AWR, LX, JDC, POW.  The paper was written by PSR with contributions from all coauthors.

        *Competing interests.* The authors declare that they have no conflict of interest.

*Acknowledgements.*  This research was funded by the ATom investigation under NASA's Earth Venture program and the FIREX-AQ investigation under NASA's Upper Atmospheric Composition Observations program (Caltech grants NNX15AG61A and 80NSSC18K0660, respectively). We would like to thank the NASA DC-8 crew and management team for support during ATom-4 and FIREX-AQ integration and flights. We thank Michelle Kim and Hannah Allen for operating Caltech's CIMS instrument during ATom-4. Data from ATom-4 are available on a
NASA online archive (https://espo.nasa.gov/atom/archive/browse/atom/id14).  Data from FIREX-AQ are available at (https://www-air.larc.nasa.gov/cgi-bin/ArcView/firexaq).

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

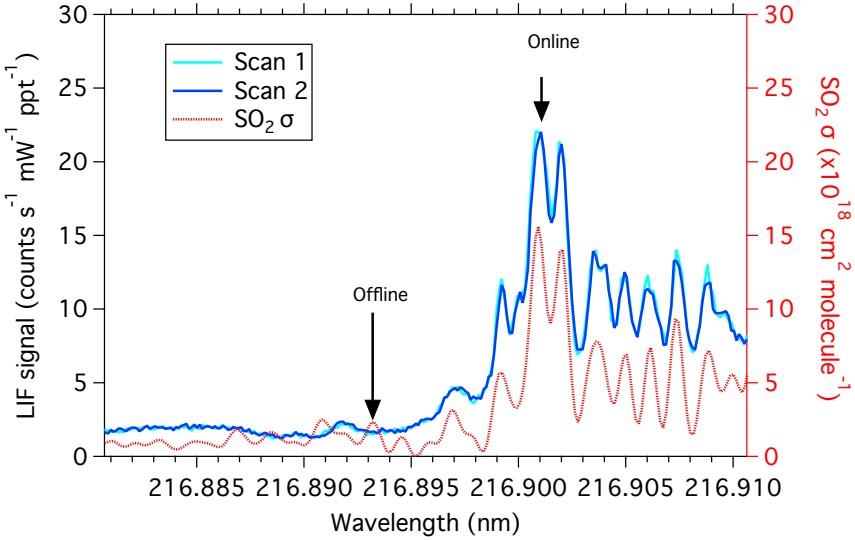

**Figure 1.** SO₂ absorption cross-section (red, right axis) in comparison to laser scans by the LIF SO₂
instrument (blue and cyan, left axis). The online wavelength is identified as the largest absorption cross-
section peak.

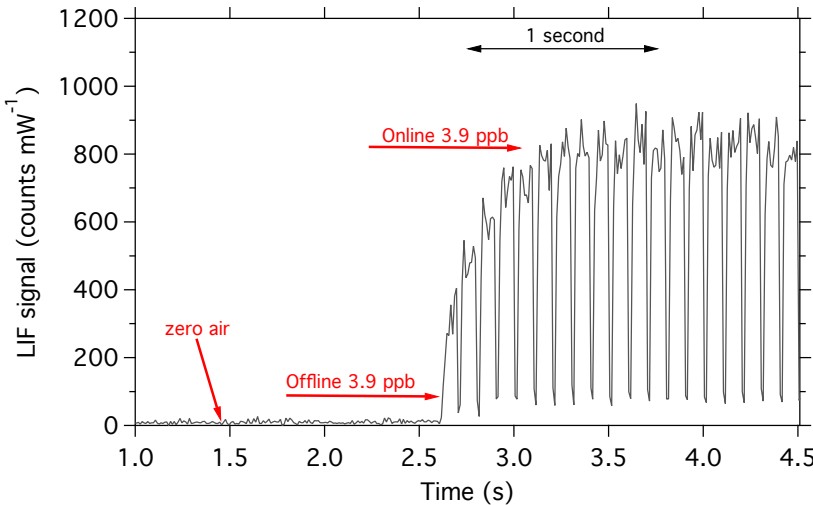

**Figure 2.** SO₂ measurements performed at 10 Hz showing a large distinction between online and offline
signals. Background measurements with zero air show the signal to be negligible in the absence of SO₂.

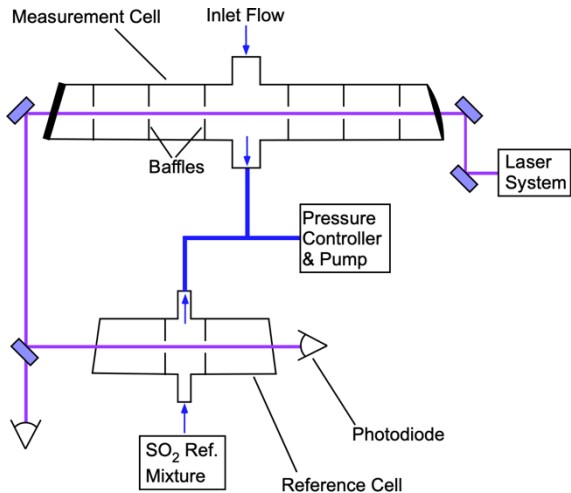

**Figure 3. LIF SO₂ detection optical bench. PMTs are located above the plane of this schematic and oriented to collect fluorescence from the center of each cell.**

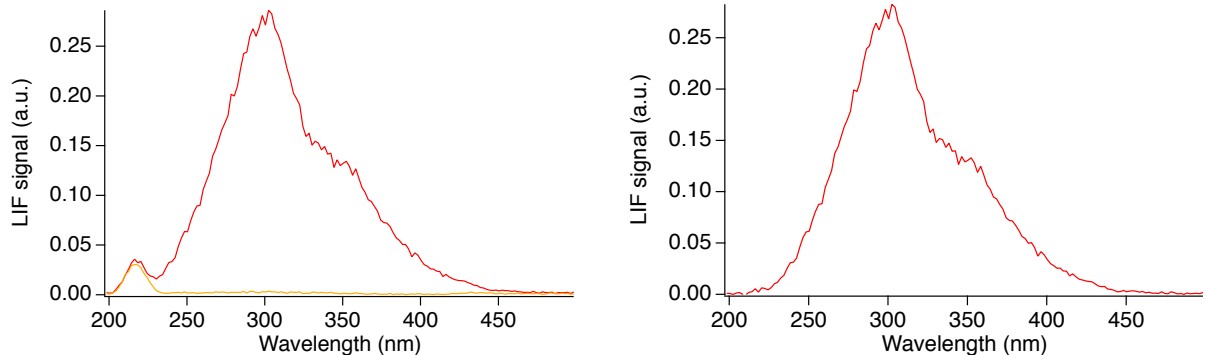

**Figure 4. Fluorescence spectrum observed from SO₂ (red) in addition to Rayleigh scattering and background (orange) (left) and the absolute SO₂ fluorescence after subtraction of the background (right).**

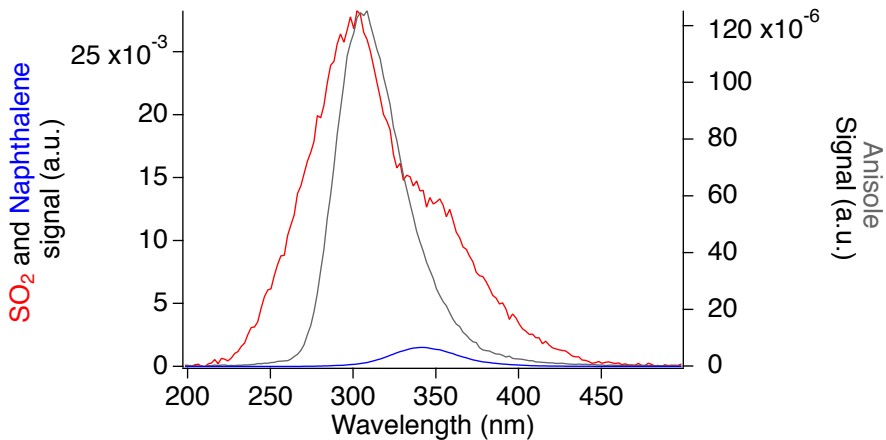

**Figure 5. Fluorescence spectra of SO₂ (red), naphthalene (blue), and anisole (grey) normalized by the calculated concentrations.**

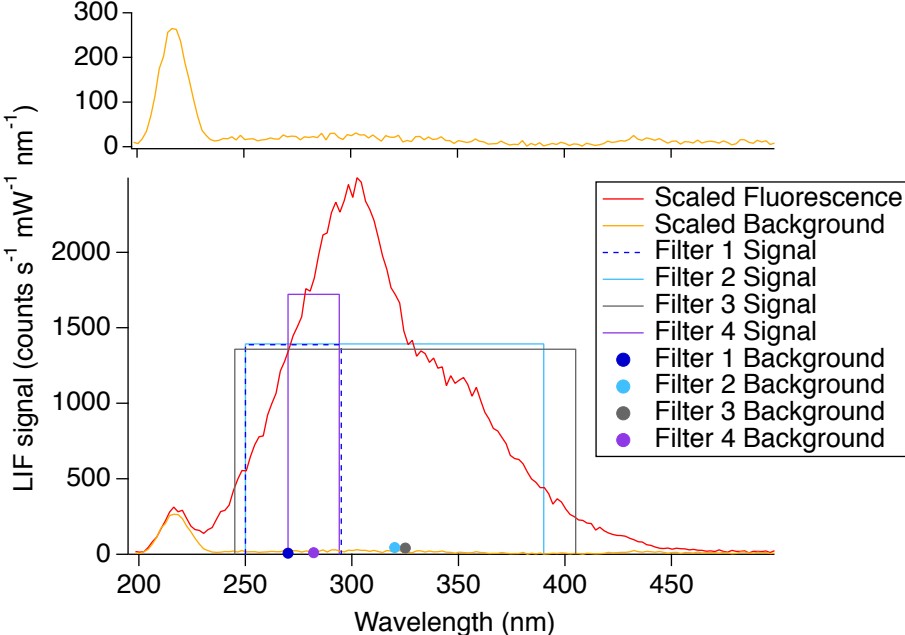

**Figure 6. Adjusted SO₂ fluorescence spectrum (red line) and background (orange line, top and bottom panels). The filter measurements (boxes and solid circles) show the filter range by the width of the box and the height indicates the observed signal. The solid circles indicate the background observed for the corresponding filter. Filter 1 is the Semrock 300/SP-25 plus the Semrock 244RS-25. Filter 2 is the Schott UG11 plus the Semrock 244RS-25. Filter 3 is the Asahi XUV0400 plus the Thorlabs FGUV5. Filter 4 is the Semrock 280/10-25 plus the Semrock 244RS-25.**

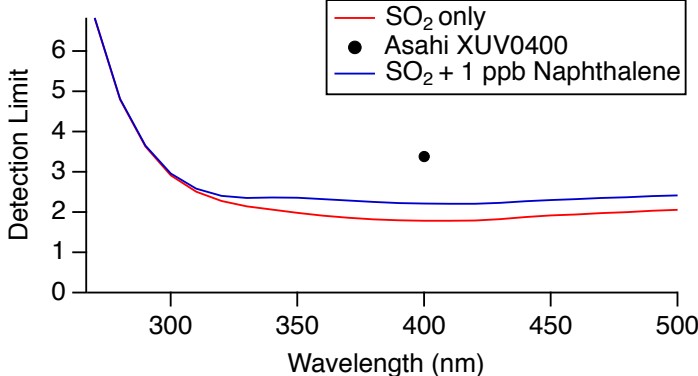

**Figure 7. Calculated detection limit of the LIF SO₂ instrument (red line), with the addition of 1 ppb naphthalene (blue line), and the measured detection limit of Asahi XUV0400 bandpass filter (black marker) with a 1 second integration period.**

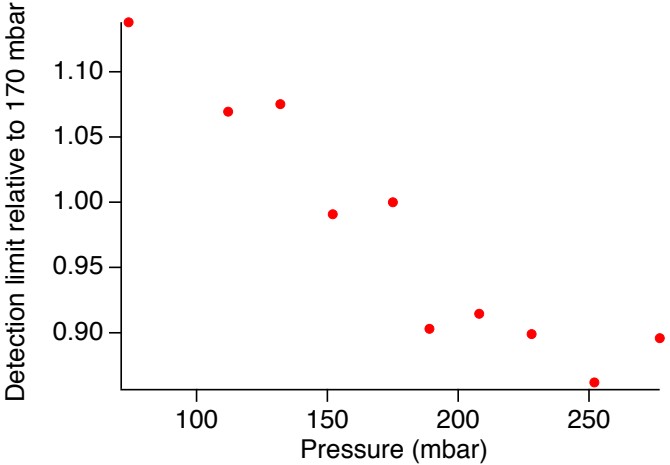

565    **Figure 8. Measured LIF SO₂ detection limit over a pressure range of 74-277 mbar.**

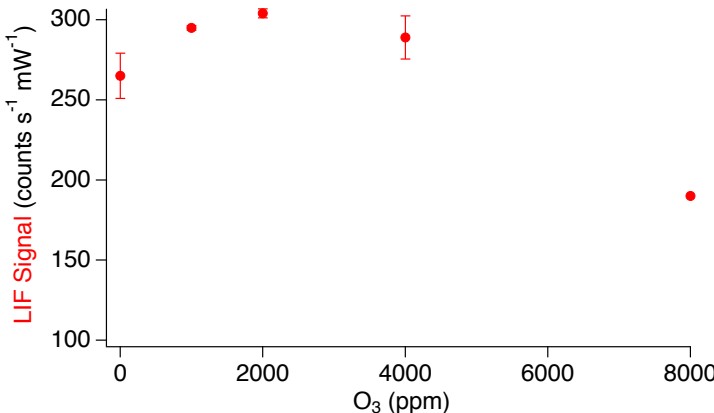

**Figure 9. LIF signal as a result of ozone addition in the range of 0-8000 ppm.**

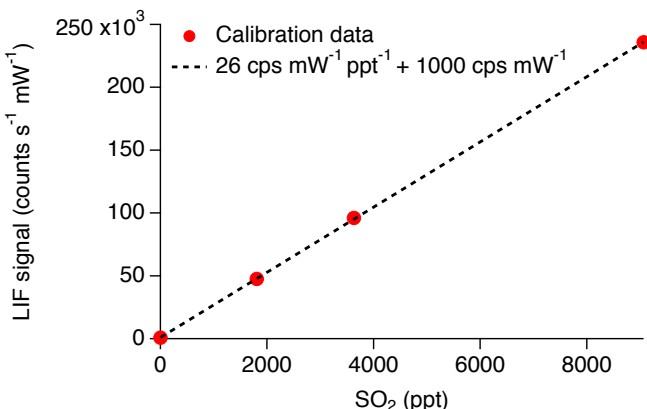

570

**Figure 10. Closed circles indicate the linearized count rate divided by measured laser power observed during calibration. Dashed line represents the fit to the calibration data indicating a sensitivity of 26 cps mW⁻¹ ppt⁻¹ and background of 1000 cps mW⁻¹.**

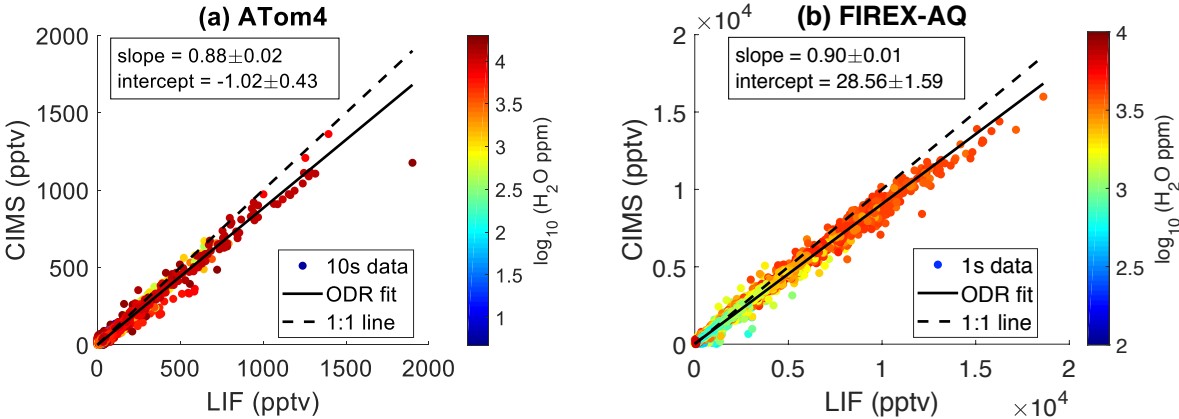

575 Figure 11.  Correlation between CIT CIMS and LIF SO₂ during (a) all 12 of the NASA ATom-4 flights and (b) one FIREX-AQ flight on 08/03/2019.  The dashed line represents the 1:1 ratio and the solid line represents the orthogonal fit. 10-second averaged data are used in the ATom-4 comparison to improve the signal/noise and 1-second averaged data are used in the FIREX-AQ comparison.