# Peer review of "Improvements to a laser-induced fluorescence instrument for measuring SO2: impact on accuracy and precision"

_Atmospheric Measurement Techniques, 2020_

## Referee Comment (RC1) · Anonymous Referee #1 · 16 Nov 2020

The authors describe the further development of an SO2 fluorescence instrument that has been described before. The improvements are of interest for the reader. Therefore, the manuscript is within the scope of AMT. However, the manuscript lacks in describing details and discussion of results and would benefit from a clearer description and discussion of results. There is little effort to bring results into the context of literature.

Detailed comments:

Line 44-64: The discussion of SO2 emissions is rather confusing. The author should be clearer in the discussion of increases and decreases of emissions specifically which time and emission sector is referred to.

[Figure]

Line 135: What is meant by "simpler design"?

Line 139: Why exactly can the system now be operated at 200kHz instead of 25kHz?

Line 144: What exactly is the reason why the laser linewidth is now narrower compared to the previous system?

Line 175: Please state the wavelengths for the stated absorption cross sections.

Line 208- 227: Does the calculation of FWHM takes the resolution of 20nm of the experimental set-up into account? Does this include the second red-shifted peak in the case of $SO_2$? Was there an effort to de-convolute the spectrum taking these effects into account?

Line 235: Because the timing of the fluorescence detection becomes important in the discussion, this should be explained in more detail in the instrument description.

Line 237-240: The authors may want to make the point that not only fluorescence is required for an interference, but also that the excitation spectrum must be similar. The author may want to consider showing excitation spectra of the aromatic compounds in addition. There is no comparison about fluorescence and excitation spectra found in literature. This should be added.

Section 4: This entire section is rather confusing. The motivation for testing different bandpass filters should be made clear. Why are there only the results from one filter shown in Fig 7? A clear discussion of Fig 7 and conclusions with respect to the different filters is missing.
* * *

---

## Referee Comment (RC2) · Anonymous Referee #2 · 31 Dec 2020

This manuscript describes improvements performed on a laser induced fluorescence instrument dedicated to airborne atmospheric measurements of SO2 to (1) improve both precision and limit of detection and (2) assess whether other fluorescent species could interfere during ambient measurements.

The work reported in this publication was carefully performed and is described in a clear and concise manner. This publication will be of interest for the scientific community. I therefore recommend publication with only a few minor comments:

L123-126: "...while precision sufficient for measurements in the UT/LS was achieved, the detection limit was determined by the background from scattered photons ..." -

[Figure]

Please indicate the precision reached on this version of the instrument and the main sources of noise contributing to the background signal.

L269-270: "Furthermore, the detection limit shows a decreasing trend with increasing pressure (Fig. 8) due to a linear increase in fluorescence signal in this regime." – When the pressure increases in the measurement cell, wouldn't we expect a reduction of the fluorescence yield due to an increase of collisional quenching. If so, why is the fluorescence signal increasing linearly with the pressure?

Section 6: It seems that calibrations were performed under dry conditions. Could the $SO_2$ fluorescence signal be impacted by quenching from ambient water vapor?

Figure 11 (right panel: FIREX-AQ): not discussed in the text

Figure 11 (insert): Please add significant figures for the uncertainty reported on the slope

---

## Author Comment (AC1) · 1 Feb 2021

**Response to the reviewers for the* "Improvements to a laser-induced fluorescence instrument for measuring SO₂: impact on accuracy and precision" study *by* Pamela S. Rickly et al.**

We thank the reviewers for their insightful comments which has helped to improve the manuscript. Below are our replies to each comment in blue.

**Anonymous Referee #1**

**Comment #1:** Line 44-64: The discussion of SO2 emissions is rather confusing. The author should be clearer in the discussion of increases and decreases of emissions specifically which time and emission sector is referred to.

**Response:** We have changed the text within the paragraph to reflect a clearer discussion of the emissions.

**Comment #2:** Line 135: What is meant by "simpler design"?

**Response:** This was meant to indicate a simpler mode of operation. Clarification has been made.

**Comment #3:** Line 139: Why exactly can the system now be operated at 200kHz instead of 25kHz?

**Response:** Previously, the lower repetition rate was used to achieve higher energy pulses and more UV laser power. Now, we have re-optimized the system at a higher repetition rate. Some additional text has been added to describe this.

**Comment #4:** Line 144: What exactly is the reason why the laser linewidth is now narrower compared to the previous system?

**Response:** As mentioned in this section, the seed laser is operated at a constant current mode instead of very short pulses. This, along with lower peak powers resulting in less spectral broadening in the fibers has reduced the laser linewidth. Some text has been added in section 2.2 to better explain this.

**Comment #5:** Line 175: Please state the wavelengths for the stated absorption cross sections.

**Response:** Done.

**Comment #6:** Line 208- 227: Does the calculation of FWHM takes the resolution of 20nm of the experimental set-up into account? Does this include the second red-shifted peak in the case of SO2? Was there an effort to de-convolute the spectrum taking these effects into account?

**Response:** We did not do any de-convolution of the spectrum. Now we explain more clearly how the observed emission spectrum is impacted by the resolution of our experiment.

**Comment #7:** Line 235: Because the timing of the fluorescence detection becomes important in the discussion, this should be explained in more detail in the instrument description.

**Response:** This has been added at Line 237-239.

**Comment #8:** Line 237-240: The authors may want to make the point that not only fluorescence is required for an interference, but also that the excitation spectrum must be similar. The author may want to consider showing excitation spectra of the aromatic compounds in addition. There is no comparison about fluorescence and excitation spectra found in literature. This should be added.

**Response:** A comment and citations have been added here.

**Comment #9:** Section 4: This entire section is rather confusing. The motivation for testing different bandpass filters should be made clear. Why are there only the results from one filter shown in Fig 7? A clear discussion of Fig 7 and conclusions with respect to the different filters is missing.

**Response:** The bandpass filter results shown in Figure 7 represent the detection limit of the bandpass filter that was found to be most optimal for measurements of SO2. This section has been edited for clarity.

**Anonymous Referee #2**

**Comment #10:** L123-126: ". . .while precision sufficient for measurements in the UT/LS was achieved, the detection limit was determined by the background from scattered photons . . ." - Please indicate the precision reached on this version of the instrument and the main sources of noise contributing to the background signal.

**Response:** This has been added.

**Comment #11:** L269-270: "Furthermore, the detection limit shows a decreasing trend with increasing pressure (Fig. 8) due to a linear increase in fluorescence signal in this regime." – When the pressure increases in the measurement cell, wouldn't we expect a reduction of the fluorescence yield due to an increase of collisional quenching. If so, why is the fluorescence signal increasing linearly with the pressure?

**Response:** The reviewer has made a good suggestion about the need to further explain the fluorescence signal trend with pressure. While collisional quenching is an important consideration for LIF detection of other molecules, the short $SO_2$ lifetime (~5 ns) limits the importance of fluorescence quenching by surrounding molecules. Instead, the fluorescence signal continues to increase linearly as the number of $SO_2$ molecules increases with pressure. This discussion has been added.

**Comment #12:**  Section 6: It seems that calibrations were performed under dry conditions. Could the SO2 fluorescence signal be impacted by quenching from ambient water vapor?

**Response:**  It is expected that water vapor will not create additional fluorescence quenching due to the short fluorescence lifetime of $SO_2$ as further explained in response to comment #11.  This was demonstrated in our prior work (Rollins et al., 2016) and is now noted in section 4.

**Comment #13:**  Figure 11 (right panel: FIREX-AQ): not discussed in the text

**Response:**  This has been added.

**Comment #14:**  Figure 11 (insert): Please add significant figures for the uncertainty reported on the slope

**Response:**  This has been added.